# Identification of Two Dysfunctional Variants in the ABCG2 Urate Transporter Associated with Pediatric-Onset of Familial Hyperuricemia and Early-Onset Gout

**DOI:** 10.3390/ijms22041935

**Published:** 2021-02-16

**Authors:** Yu Toyoda, Kateřina Pavelcová, Jana Bohatá, Pavel Ješina, Yu Kubota, Hiroshi Suzuki, Tappei Takada, Blanka Stiburkova

**Affiliations:** 1Department of Pharmacy, The University of Tokyo Hospital, Tokyo 113-8655, Japan; ytoyoda-tky@umin.ac.jp (Y.T.); yukubota-tky@umin.ac.jp (Y.K.); suzukihi-tky@umin.ac.jp (H.S.); tappei-tky@umin.ac.jp (T.T.); 2Institute of Rheumatology, 128 00 Prague, Czech Republic; pavelcova@revma.cz (K.P.); bohata@revma.cz (J.B.); 3Department of Rheumatology, First Faculty of Medicine, Charles University, 121 08 Prague, Czech Republic; 4Department of Pediatrics and Inherited Metabolic Disorders, First Faculty of Medicine, Charles University and General University Hospital, 121 00 Prague, Czech Republic; pavel.jesina@vfn.cz

**Keywords:** *ABCG2* genotype, clinico-genetic analysis, ethnic specificity, genetic variations, precision medicine, rare variant, Roma, serum uric acid, SUA-lowering therapy, urate transporter

## Abstract

The *ABCG2* gene is a well-established hyperuricemia/gout risk locus encoding a urate transporter that plays a crucial role in renal and intestinal urate excretion. Hitherto, p.Q141K—a common variant of ABCG2 exhibiting approximately one half the cellular function compared to the wild-type—has been reportedly associated with early-onset gout in some populations. However, compared with adult-onset gout, little clinical information is available regarding the association of other uricemia-associated genetic variations with early-onset gout; the latent involvement of ABCG2 in the development of this disease requires further evidence. We describe a representative case of familial pediatric-onset hyperuricemia and early-onset gout associated with a dysfunctional ABCG2, i.e., a clinical history of three generations of one Czech family with biochemical and molecular genetic findings. Hyperuricemia was defined as serum uric acid (SUA) concentrations 420 μmol/L for men or 360 μmol/L for women and children under 15 years on two measurements, performed at least four weeks apart. The proband was a 12-year-old girl of Roma ethnicity, whose SUA concentrations were 397–405 µmol/L. Sequencing analyses focusing on the coding region of *ABCG2* identified two rare mutations—c.393G>T (p.M131I) and c.706C>T (p.R236X). Segregation analysis revealed a plausible link between these mutations and hyperuricemia and the gout phenotype in family relatives. Functional studies revealed that p.M131I and p.R236X were functionally deficient and null, respectively. Our findings illustrate why genetic factors affecting ABCG2 function should be routinely considered in clinical practice as part of a hyperuricemia/gout diagnosis, especially in pediatric-onset patients with a strong family history.

## 1. Introduction

Serum urate concentration is a complex phenotype influenced by both genetic and environmental factors, as well as interactions between them. Hyperuricemia results from an imbalance between endogenous production and excretion of urate. This disorder is a central feature in the pathogenesis of gout [1], which progresses through several degrees, i.e., asymptomatic hyperuricemia, acute gouty arthritis, intercritical gout, and chronic tophaceous gout. While not all individuals with hyperuricemia develop symptomatic gout, the risk of gout increases in proportion to the elevation of urate in circulation. In addition to hyperuricemia, the risk is also associated with gender, weight, age, environmental, and genetic factors [2,3], and interactions between them all. Recent data suggest that the number of gout patients under the age of 40 years (early-onset) is increasing [4]. These early-onset patients may have different clinical signs and co-morbidities from those who present with gout at a later age [5,6]. Given the development of earlier metabolic disorders in the early-onset gout patients compared with common gout patients [5], together with the need for continuous management of health from their younger age, understanding the risks of early-onset gout is clinically important.

More and more evidence suggests that the net amount of excreted uric acid is mainly regulated by physiologically important urate transporters, such as urate transporter 1 (URAT1, known as SLC22A12, a renal urate re-absorber) [7], glucose transporter member 9 (GLUT9, also known as SLC2A9) [8,9], and ATP-binding cassette transporter G2 (ABCG2, a high capacity urate exporter expressed in the kidneys and intestines) [10,11,12,13]. Dysfunction of URAT1 and GLUT9 reportedly cause inherited hypouricemia type 1 and type 2, respectively, while dysfunction of ABCG2 is a risk factor for hyperuricemia and gout [1,14]. Additionally, the ABCG2 population-attributable percent risk for hyperuricemia has been reported to be 29.2%, which is much higher than for those with more typical environmental risks such as BMI ≥ 25.0 (18.7%), heavy drinking (15.4%), and age ≥ 60 years old (5.74%) [15]. Hence, dysfunctional variants of ABCG2 may affect clinical outcomes by influencing the accumulation of uric acid in the body.

The ABCG2 protein, which is an *N*-linked glycoprotein composed of 655-amino acid, is a homodimer membrane transporter found in a variety of tissues [16,17,18]. ABCG2 is expressed on the brush border membranes of renal and intestinal epithelial cells, where ABCG2 is involved in the ATP-dependent excretion of numerous substrates from the cytosol into the extracellular space. ABCG2 was historically first described as a drug transporter linked to breast cancer resistance [19,20,21], which led to many studies that focused on its critical role in drug pharmacokinetics. To date, not only xenobiotics but also endogenous substances, including uremic toxin [22] and urate [11,12], have been identified as ABCG2 substrates.

In the context of hyperuricemia/gout, there are about 50 allelic variants, including a number of rare variants with minor allele frequencies (MAF) < 0.01%, which have been found in the *ABCG2* gene. Wide ethnic differences have been found relative to the frequencies of these alleles. There are two well-studied, common ABCG2 allelic variants—p.V12M (c.34G>A, rs2231137) and p.Q141K (c.421C>A, rs2231142) that have highly variable frequencies depending on ethnicity. Both are commonly found in Asians (in a relatively large number of ethnic groups) but are rarely found in Europeans [23]. A minor allele of p.V12M appears to be protective regarding susceptibility to gout [24]; however, this apparent effect is due to linkage disequilibrium between p.V12M and other dysfunctional ABCG2 variants [25]. In other words, the V12M mutation has little impact on the function of ABCG2. On the other hand, the p.Q141K variant decreases ABCG2 levels, which reduces the cellular function of ABCG2, as a urate exporter, by 50% [12]. In addition to p.Q141K, p.Q126X (c.376C>T, rs72552713), which is common in the Japanese population (MAF, 2.8%) [26] but rare in other populations, has been identified as hyperuricemia- and gout-risk allele [12]. Given that p.Q141K and p.Q126X variants are both associated with a significantly increased risk of gout, the effects of dysfunctional variants of ABCG2, relative to gout susceptibility, are genetically strong [27].

In a recent study focusing on 10 single nucleotide polymorphisms in 10 genes (*ABCG2*, *GLUT9*/*SLC2A9*, *SLC17A1*, *SLC16A9*, *GCKR*, *SLC22A11*, *INHBC*, *RREB1*, *PDZK1*, and *NRXN2*) that are strongly associated with serum uric acid (SUA) concentrations, only ABCG2 p.Q141K was associated with early-onset gout (< 40 years of age) in European and Polynesian subjects [28]. Additionally, in a previous study, we found that the MAF of p.Q141K in a cohort of hyperuricemia and gout with pediatric-onset was 38.7%, which was significantly higher than adult-onset (21.2%) as well as normouricemic controls (8.5%) [29]. This information suggests that ABCG2 dysfunction could be strongly associated with pediatric-onset hyperuricemia and early-onset gout; however, compared with adult-onset gout, little clinical information is available, except for the ABCG2 p.Q141K, regarding early-onset gout linked to other SUA-associated mutations. Furthermore, the latent involvement of ABCG2 in the development of this disease requires further evidence.

In this study, we investigated the genetic cause of pediatric-onset hyperuricemia and early-onset gout over three generations of a single family. Based on a positive family history of hyperuricemia or gout, we identified two rare mutations, c.393G>T (p.M131I) and c.706C>T (p.R236X), in the *ABCG2* gene. A series of biochemical assays revealed that ABCG2 p.M131I and p.R236X were functionally deficient and null, respectively. Our results also contributed to a more in-depth understanding of the effects of rare *ABCG2* variants, which is a highly polymorphic gene, on its function as a physiologically important transporter.

## 2. Results

### 2.1. Subjects

The clinical and biochemical data from this study are summarized in Table 1. Repeated biochemical analysis of the proband (a 12-year-old girl with chronic asymptomatic hyperuricemia) showed elevated serum urate (397–405 μmol/L; 6.67–6.81 mg/dL) with a decreased fractional excretion of uric acid (FE-UA) (2.2–3.5%). No clinical or laboratory symptoms of renal disease were present in the patient. The girl’s mother (38-years-old) had previously presented with asymptomatic hyperuricemia (420–439 μmol/L; 7.06–7.38 mg/dL of serum urate); both the maternal grandfather (66-years-old) and the maternal uncle (40-years-old) had presented with hyperuricemia (500–537 µmol/L; 8.41–9.03 mg/dL) and early-onset gout, with the first gout attack occurring in the uncle when he was 35-years-old. The girl’s father (38-years-old) presented with asymptomatic hyperuricemia (487 μmol/L; 8.19 mg/dL of serum urate), which seemed to be associated with metabolic syndrome (i.e., hypertriglyceridemia, hypertension, and central obesity). Metabolic investigation for purine metabolism (i.e., hypoxanthine and xanthine levels in the urine) found normal urinary excretion, suggesting that hyperuricemia in our patients and family relatives was not caused by an excess production of uric acid. Thus, we focused on the physiologically important urate transporters, especially ABCG2, which regulate urate handling in the body as described below.

### 2.2. Genetic Analysis

Targeted exon sequencing analyses of *ABCG2* revealed that the proband had two rare heterozygous non-synonymous mutations, i.e., (1) c.393G>T (p.M131I, rs759726272) and (2) c.706C>T (p.R236X, rs140207606). No other exonic mutations were found, including previously characterized common genetic risk factors for hyperuricemia/gout, such as *ABCG2* c.376C>T (p.Q126X) and c.421C>A (p.Q141K). Moreover, a segregation analysis of the family confirmed the two mutations as probably being disease-associated mutations. Heterozygous c. 393G>T and c.706C>T were identified through the maternal line in the mother, uncle, and grandfather of the proband. The pedigree is shown in Figure 1 along with representative electropherograms of partial *ABCG2* sequences showing a heterozygous missense mutation (c.393G>T) in exon 5 and a nonsense mutation (c.706C>T) in exon 7. Moreover, sequencing analyses, including analyses of previously known SUA-associating loci, found no disease-associated non-synonymous mutations in *GLUT9* and *URAT1*. Although two mutations, c.73G>A (p.G25R, rs2276961) and c.844G>A (p.V282I, rs16890979), were found in *GLUT9*; however, previous studies have shown that they would not cause hyperuricemia/gout [30,31].

Further sequencing analyses revealed that neither of these *ABCG2* mutations was detected in a previously enrolled cohort of 360 patients with hyperuricemia/gout and a cohort of 132 normouricemia control subjects of European descendent [32]. However, in another control group of 60 subjects of Roma ethnicity [33], although serum urate and FE-UA values were not available, we found one copy of c.393G>T (p.M131I), but without c.706C>T (p.R236X), in a male hemodialysis patient with severe chronic kidney disease. This finding suggested that c.393G>T (p.M131I) and c.706C>T (p.R236X) can occur independently, which led us to examine the effect of each non-synonymous mutation on the ABCG2 protein.

Additionally, the independent genetic origin of these two *ABCG2* mutations was also strongly supported by supplementary analyses of MAF data that is openly available from the Exome Aggregation Consortium (http://exac.broadinstitute.org/ (accessed on 30 November 2020)). With regard to c.393G>T (p.M131I), only six heterozygotes of the minor allele (393T) were found in a cohort of 64,397 European subjects (MAF, 0.0047%); with even fewer carriers in other populations (total worldwide MAF frequency = 0.0021%). With regard to c.706C>T (p.R236X), 48 heterozygotes of the minor allele (706T) were found in a cohort of 645,050 European subjects (MAF = 0.0372%, which is almost 8 times greater than the MAF of c.393G>T); in other populations, frequencies ranging from 0.0000% in European Finnish to 0.1254% (the highest) in Ashkenazi Jewish were found.

### 2.3. Functional Analysis

To investigate the effect of the two identified non-synonymous mutations (p.M131I and p.R236X) on the intracellular processing and function of the ABCG2 protein, we conducted several biochemical analyses using transiently ABCG2-expressing mammalian cells (Figure 2). First, immunoblotting for *N*-glycosidase (PNGase F)-treated whole cell lysates (Figure 2A) demonstrated that p.M131I had a minimal effect on protein levels and *N*-linked glycosylation status, while the p.R236X variant was detected as truncated forms of the protein with weaker band intensities compared to ABCG2 wild-type (WT), as would be expected based on its amino acid sequence. Confocal microscopy (Figure 2B) showed that, like ABCG2 WT, the p.M131I variant was mainly located on the plasma membrane, while the p.R236X variant exhibited little plasma membrane localization. Expression of the p.M131I variant as a glycoprotein on plasma membrane-derived vesicles was confirmed using immunoblotting (Figure 2C). Next, using a vesicle transport assay [34] with an optimized experimental procedure for determining the initial rate of urate transport by ABCG2 based on our previous studies [12,35], ABCG2 function was evaluated as having ATP-dependent urate transport activity into plasma membrane vesicles (Figure 2D). The functional assay revealed that contrary to the WT, p.M131 had limited ATP-dependent urate transport activity, with the ABCG2-mediated urate transport activity of this variant calculated to be 14 ± 2% of the WT controls. The p.R236X variant was functionally null (Figure 2D), which was consistent with the results of our biochemical analyses (Figure 2A–C).

## 3. Discussion

In this study, we found and analyzed a representative case of pediatric-onset hyperuricemia and early-onset gout in a Czech family associated with two newly identified, one functionally deficient and the other a null mutation, in *ABCG2*. A positive family history of hyperuricemia/gout in the context of ABCG2 dysfunction was observed in the maternal line (the mother, maternal uncle, and maternal grandfather) of the proband (a young girl, of Roma ethnicity, with chronic asymptomatic hyperuricemia); her father exhibited elevated serum urate that seemed to be associated with metabolic syndrome. Although a decrease in net renal urate excretion was observed in all cases of hyperuricemia in this family, which was characterized by FE-UA < 5%, our findings suggest that the hyperuricemia was linked to heterogeneity. Moreover, the familial hyperuricemia/gout observed in this study was not associated with a common variant, such as p.Q141K, but with a rare ABCG2 variant; this supports the recently proposed genetic concept, i.e., the “Common Disease, Multiple Common and Rare Variant” model [25,36], for the association between hyperuricemia/gout and the *ABCG2* gene.

Our findings of ABCG2 variants will provide deeper insights into amino acid positions that are critical for normal ABCG2 function. Based on cryo-electron microscopy (cryo-EM) of ABCG2 [37], both the M131 and R236 residues are in the cytoplasmic region of the N-terminus of the ABCG2 protein. Regarding p.M131I, the original amino acid (M131) sequence is conserved in several major mammalian species (Figure 3). Unlike the p.Q141K variant, which affects intracellular processing of the ABCG2 protein [38], the p.M131I variant disrupts ABCG2’s function as a urate transporter, with little effect on the ABCG2 protein or its cellular localization. A plausible explanation for the molecular mechanism of the p.M131I effect is that this amino acid substitution could affect ABCG2 substrate specificity and/or affect its ATPase activity, including the binding affinity of ATP, which is the driving force for ABC transporters. The latter possibility is supported by a structural feature that the M131 residue is located near the conserved glutamine (Q126) within a Q-loop in the nucleotide-binding domain of ABCG2, a key element for the catalytic cycle of ATP binding and hydrolysis [37,39]. Further studies are needed to address this biochemical issue, which may deepen the mechanistic insight of ABCG2 protein. With regard to p.R236X, the acquired stop codon results in the production of a shortened ABCG2 variant (only about one-third the amino acid length of the native ABCG2 protein) that lacks all the transmembrane domains essential for normal protein function as a membrane transporter. This is consistent with our results demonstrating that the p.R236X variant is functionally null. Thus, the c.[393G>T; c.706C>T] (p.[M131I; R236X]) variant does not function as a urate transporter.

To discuss the expected independence in the genetic origin of two *ABCG2* mutations identified in this study, we will focus on the genetic specificity of each population. The Czech family studied in this study is of Roma ethnicity. The Roma composes a transnational ethnic population of 8–10 million, with the original homeland being India; currently, they are the largest and the most widespread ethnic minority in Europe. The founder effect and subsequent genetic isolation of the Roma have led to a population specificity regarding the genetic background of specific human diseases. In other words, mutations associated with rare diseases found in the Roma population tend to be at extremely low frequencies in other European populations, and vice versa. Indeed, several genetic variants causing rare diseases are unique to the Roma, and many of these mutations have only recently been discovered, e.g., Charcot Marie Tooth disease type 4D and 4G (OMIM 601455 and 605285), Congenital cataract facial dysmorphism neuropathy (OMIM 604166), Gitelman syndrome (OMIM 263800), and Galactokinase deficiency (OMIM 230200) [41,42]. In light of the genetic specificity found in the Roma, we investigated the frequency of two identified *ABCG2* mutants (c.393G>T and c.706C>T) in our control cohort of 60 subjects of Roma origin (see Section 2.2 in Results). MAFs of c.393G>T in the Roma cohort and European population were 0.833% (1 allele/120 subjects) and 0.002% (6 alleles/64,397 subjects), respectively. Although the sample size of our control cohort was very modest, it could be large enough to imply that the origin of c.393G>T might be the Roma population. On the other hand, the prevalence of c.706C>T was higher in Europeans than in the Roma, suggesting that these two *ABCG2* mutations could have different genetic origins.

For clinical practice, our findings suggest a need for further discussion about the potential benefits of urate-lowering therapy after a diagnosis of hyperuricemia in pediatric-onset patients with ABCG2 dysfunction. Interestingly, harboring p.Q141K is reportedly associated with inadequate response to allopurinol (characterized by a smaller reduction in serum urate concentrations compared with WT) [29,43,44,45]; allopurinol, which inhibits uric acid production, is a well-known and widely used drug for lowering SUA. Although the mechanisms of action for the inadequate response are still unclear, other SUA-lowering drugs may be somewhat better in terms of efficacy for patients with a dysfunctional *ABCG2* allele, which puts them at higher risk of developing hyperuricemia/gout. On the other hand, among the clinically-used inhibitors for the production of uric acid (i.e., allopurinol, febuxostat, and topiroxostat), only febuxostat, to the best of our knowledge, strongly and clinically inhibits ABCG2 function as a urate transporter [35]. This is also supported by a recent clinical study that showed orally-administered febuxostat inhibits intestinal ABCG2 in humans [46]. In this context, the efficacy of febuxostat as an SUA-lowering drug can be partially blocked by ABCG2 inhibition, except in cases of completely null ABCG2 function. Considering this complexity, as well as risks of adverse effects [47], the best SUA-lowering drugs, including uricosuric agents and uric acid production inhibitors, should be carefully determined for each patient.

Additionally, we can emphasize the clinical importance of the documentation regarding the family we addressed in this study, given the infrequency of detailed studies on SUA levels in children. Indeed, pediatric-onset of hyperuricemia is relatively rare in clinical practice; it is often associated with rare conditions (e.g., purine metabolic disorders; kidney disorders including uromodulin-associated disorders; metabolic genetic disorders including glycogen storage disease, hereditary fructose intolerance, and mitochondrial disorders). As we showed previously [48], the levels of SUA and FE-UA are quite dynamic in the first year of life. In brief, the SUA levels are low in infancy (131–149 μmol/L; 2.2–2.5 mg/dL at 2–3 months of age) due to the high FE-UA levels (>10%); the FE-UA levels decrease to approximately 8% at age 1 and then stay through childhood, which is associated with mean SUA levels (208–268 μmol/L; 3.5–4.5 mg/dL) of children. At adolescence (after age 12), the FE-UA levels significantly decrease in boys but not in girls, resulting in a further significant increase in SUA levels in young men but not in young women. These pieces of information support the rarity of our case of which proband is a 12-year-old girl with chronic asymptomatic hyperuricemia characterized by elevated serum urate (397–405 μmol/L; 6.67–6.81 mg/dL) and the decreased FE-UA (2.2–3.5%).

Some limitations warrant mention. It is unclear whether there could have been other genetic factors also affecting the early-onset phenotypes in the family we studied, although harboring dysfunctional *ABCG2* mutations is the most plausible explanation, as the study showed. A previous study showed that in the context of extra-renal underexcretion hyperuricemia, a genetic dysfunction of ABCG2 increased SUA levels and apparent urinary urate excretion, which was coupled with decreased intestinal urate excretion. Thus, we can assume the presence of latent mechanisms causing the decreased (<5%) FE-UA levels observed in our patients. Although there is little information available regarding this, a possible factor could be increased renal urate reabsorption, which can be mediated by up-regulation of renal urate re-absorbers, including *URAT1/SLC22A12, GLUT9*/*SLC2A9*, and *organic anion transporter 10* (known as *SLC22A13*) [49]. In this context, genetic variations affecting the expression of such genes will be the targets of future studies.

In conclusion, we found a representative case of pediatric hyperuricemia with familial gout that harbored two dysfunctional *ABCG2* mutations. Genetic variations in *ABCG2* should be kept in mind during diagnostic procedures for pediatric-onset hyperuricemia. Considering *ABCG2* genotypes will be beneficial for patients with early-onset and/or familial hyperuricemia and gout. This type of genetic information will also allow a personalized approach regarding the best urate-lowering treatment (i.e., uric acid production inhibitors or uricosuric agents) for patients with dysfunctional ABCG2 variants, as well as the best time to initiate pharmacotherapy for hyperuricemia.

## 4. Materials and Methods

### 4.1. Clinical Subjects 

The studied proband and her family members were Czechs of Roma ethnicity diagnosed with familial (early-onset) hyperuricemia/gout. Written informed consent was obtained from each subject before enrollment in the study. All tests were performed in accordance with standards set by the institutional ethics committees, which approved 30 June 2015 the project no. 6181/2015. All the procedures were performed in accordance with the Declaration of Helsinki.

Hyperuricemia was defined as serum urate levels greater than 420 μmol/L (7.06 mg/dL) for men or 360 μmol/L (6.05 mg/dL) for women and children under 15 years on two measurements, performed at least four weeks apart. Gouty arthritis was diagnosed according to the American College of Rheumatology criteria, as follows: (1) the presence of sodium urate crystals seen in synovial fluid (using a polarized microscope, Nikon Eclipse E200, Tokyo, Japan) or (2) at least six of 12 clinical criteria being met [50].

The proband was a 12-year-old girl with a complicated perinatal anamnesis. She was born at 31 weeks of gestation with an Apgar score of 4-7-8, a birth weight of 1690 g, and a birth length of 40 cm; additionally, she developed early asphyxia syndrome. She also experienced repeated respiratory infections and was later diagnosed with bronchial asthma. She is also under the care of an ophthalmologist for myopia and astigmatism; she was also investigated for sudden onset mild bilateral cortical cataracts. She was obese (BMI = 27); her psychomotor development corresponded to her age.

### 4.2. Clinical Investigations and Sequence Analyses

Urate and creatinine levels were measured as described previously [51] using a specific enzymatic method and the Jaffé reaction, which was adapted for an auto-analyzer (Hitachi Automatic Analyzer 902; Roche, Basel, Switzerland). Metabolic investigations of purine metabolism (hypoxanthine and xanthine levels in urine) were also conducted using a method established in a previous study [51]. The proband was screened for metabolic (e.g., glycogen storage disease, hereditary fructose intolerance, and mitochondrial disorders) and kidney disease associated with hyperuricemia (e.g., uromodulin-associated disorders), using our previously published diagnostic algorithm [29].

All coding regions and exon-intron boundaries in *ABCG2* and *GLUT9/SLC2A9,* and exons 7 and 9 in *URAT1/SLC22A12* were analyzed from genomic DNA, as described previously [29,33,52]. The reference sequence for *ABCG2* was defined as version ENST00000237612.7 (location: Chromosome 4: 88,090,269–88,158,912 reverse strand) (www.ensembl.org (accessed on 30 November 2020)). For *GLUT9/SLC2A9* (NM_020041.2; NP_064425.2; SNP source dbSNP 132) and *URAT1/SLC22A12* (NM_144585.3), the reference genomic sequence was defined as version NC_000004.12 (Chromosome 4: 9,771,153–10,054,936) and NC_000011.8 (Chromosome 11: 64,114,688–64,126,396), respectively.

It is worth special mention that the world’s highest frequency of the main dysfunctional variants of URAT1, p.T467M (MAF, 5.6%) and p.L415_G417del (MAF, 1.9%), was recently identified in a Roma population (1,016 individuals) from specific regions of the Czech Republic, Slovakia, and Spain [33,53]. According to MAF data from the Exome Aggregation Consortium, p.T467M (rs200104135) showed only one heterozygous allele in a cohort of 15,296 in a South Asian population (MAF, 0.003%) and no occurrence in other ethnic populations; no occurrence of the p.L415_G417del allele was seen in the whole population, which supports the Roma-specific prevalence of these two URAT1 variants. For this reason, we looked for both URAT1 variants and confirmed that neither variant was present in our studied family.

### 4.3. Materials

ATP, AMP, creatine phosphate disodium salt tetrahydrate, and creatine phosphokinase type I from rabbit muscle were purchased from Sigma-Aldrich (St. Louis, MO, USA), and [8-^14^C]-uric acid (55 mCi/mmol) was purchased from American Radiolabeled Chemicals (St. Louis, MO, USA). All other chemicals were commercially available and of analytical grade.

### 4.4. Preparation of ABCG2 Variants Expression Vector

To express human ABCG2 (NM_004827.3) fused with EGFP at its N-terminus (EGFP-ABCG2) and EGFP (control), we used an ABCG2/pEGFP-C1 plasmid (open reading frame of ABCG2 was inserted into the *Hind*III and the *Apa* I sites of a pEGFP-C1 vector plasmid) that was from our previous study [32]. Of note, the functionality and expression of such construct were confirmed by previous studies we and other groups conducted [32,36,54,55,56]. Using a site-directed mutagenesis technique, the ABCG2 p.M131I (c.393G>T)/pEGFP-C1 plasmid and the ABCG2 R236X (c.706C>T)/pEGFP-C1 plasmid were generated from an ABCG2 WT/pEGFP-C1 plasmid, respectively. The introduction of each mutation was confirmed by full sequencing using BigDye Terminator v3.1 (Applied Biosystems, Foster City, CA, USA) and an Applied Biosystems 3130 Genetic Analyzer (Applied Biosystems), as described previously [40].

### 4.5. Cell Culture and Transfection

Human embryonic kidney 293 cell-derived 293A cells were purchased from Life Technologies (Carlsbad, CA, USA) and cultured in Dulbecco’s Modified Eagle’s Medium (DMEM; Nacalai Tesque, Kyoto, Japan) supplemented with 10% fetal bovine serum (Life Technologies), 1% penicillin/streptomycin, 2 mM L-glutamine (Nacalai Tesque), and 1 × Non-Essential Amino Acid (Life Technologies) at 37 °C in an atmosphere of 5% CO_2_. Each vector plasmid for ABCG2 WT, p.M131I, or p.R236X was transfected into 293A cells by using polyethyleneimine MAX (PEI-MAX; 1 mg/mL in milli-Q water, pH 7.0; Polysciences, Warrington, PA, USA) as described previously [57]. The amount of plasmid DNA used for transfection was adjusted per sample group. 

### 4.6. Preparation of Whole-Cell Lysates

Forty-eight hours after transfection, whole-cell lysates were prepared in ice-cold lysis buffer A containing 50 mM Tris/HCl (pH 7.4), 1 mM dithiothreitol, 1% (w/v) Triton X-100, and a protease inhibitor cocktail for general use (Nacalai Tesque) as described previously [58]. The protein concentration of the whole cell lysate was quantified using a BCA Protein Assay Kit (Pierce, Rockford, IL, USA) with bovine serum albumin (BSA) as a standard according to the manufacturer’s protocol. Before glycosidase treatment, the whole cell lysate samples were incubated with PNGase F (New England Biolabs Japan, Tokyo, Japan) (1.25 U/μg of protein) at 37 °C for 10 min as described previously [59], and then subjected to immunoblotting.

### 4.7. Preparation of ABCG2-Expressing Plasma Membrane Vesicles

Plasma membrane vesicles were prepared from ABCG2-expressing 293A cells, as described previously [36]. The resulting plasma membrane vesicles were rapidly frozen in liquid N_2_ and kept at −80 °C until used. The protein concentration of the plasma membrane vesicles was measured using a BCA Protein Assay Kit, as described above.

### 4.8. Immunoblotting

Expression of ABCG2 protein in whole-cell lysates and plasma membrane vesicles was assessed using immunoblotting as described previously [36], with minor modifications. In brief, the prepared samples were mixed with a sodium dodecyl sulfate-polyacrylamide gel electrophoresis sample buffer solution containing 10% 2-mercaptoethanol, separated by electrophoresis on polyacrylamide gels, and then transferred to Polyvinylidene Difluoride membranes (Immobilon; Millipore, Billerica, MA, USA) by electroblotting at 15 V for 60 min. For blocking, the membrane was incubated in Tris-buffered saline containing 0.05% Tween 20 and 3% BSA (Nacalai Tesque) (TBST-3% BSA). After overnight incubation at room temperature, blots were probed with rabbit anti-EGFP polyclonal antibodies (A11122; Life Technologies; diluted 1,500 fold in TBST-0.1% BSA), a rabbit anti-α-tubulin antibodies (ab15246; Abcam, Cambridge, MA, USA; diluted 1,000 fold), or a rabbit anti-Na^+^/K^+^-ATPase α antibodies (sc-28800; Santa Cruz Biotechnology, Santa Cruz, CA, USA; diluted 1,000 fold) followed by incubation with a donkey anti-rabbit immunoglobulin G (IgG)-horseradish peroxidase (HRP)-conjugated antibody (NA934V; diluted 4,000 fold for EGFP-ABCG2 or 3,000 fold for α-tubulin and Na^+^/K^+^-ATPase). HRP-dependent luminescence was developed using ECL^TM^ Prime Western Blotting Detection Reagent (GE Healthcare UK, Buckinghamshire, UK) and detected using a multi-imaging Fusion Solo 4^TM^ analyzer system (Vilber Lourmat, Eberhardzell, Germany).

### 4.9. Confocal Laser Scanning Microscopic Observation

For confocal laser scanning microscopy, 48 h after transfection, 293A cells were fixed with ice-cold methanol for 10 min, and then the nuclei were visualized with TO-PRO-3 Iodide (Molecular Probes, Eugene, OR, USA) as described previously [36]. To analyze the localization of EGFP-fused ABCG2 protein, fluorescence was detected using an FV10i Confocal Laser Scanning Microscope (Olympus, Tokyo, Japan).

### 4.10. Urate Transport Assay

The urate transport assay, with ABCG2-expressing plasma membrane vesicles, was conducted using a rapid filtration technique described in our previous studies [27,36], with some minor modifications. In brief, each plasma membrane vesicle (0.5 mg/mL) was incubated with 20 μM of radiolabeled urate in a reaction mixture (total 20 μL: 10 mM Tris/HCl, 250 mM sucrose, 10 mM MgCl_2_, 10 mM creatine phosphate, 1 mg/ml creatine phosphokinase, pH 7.4, and 50 mM ATP, or AMP as an ATP substitute) for 10 min at 37 °C. After incubation, the reaction mixture was mixed with 980 μL of ice-cold stop buffer (2 mM EDTA, 0.25 M sucrose, 0.1 M NaCl, 10 mM Tris-HCl, at a of pH 7.4); the resulting solution was rapidly filtered through an MF-Millipore Membrane (HAWP02500; 0.45 µm pore size and 25 mm diameter; Millipore). After washing with 5 ml of ice-cold stop buffer five times, the plasma membrane vesicles on the filter were dissolved in Clear-sol II (Nacalai Tesque). Then, the radioactivity in the plasma membrane vesicles was measured using a liquid scintillator (Tri-Carb 3110TR; PerkinElmer, Waltham, MA, USA).

The urate transport activity was calculated as the incorporated clearance (μL/mg protein/min) defined as the incorporated level of urate [disintegrations per minute (DPM)/mg protein/min]/urate level in the incubation mixture [DPM/μL]. ATP-dependent urate transport was calculated by subtracting the urate transport activity in the absence of ATP from that in the presence of ATP; ABCG2-mediated urate transport activity was calculated by subtracting ATP-dependent urate transport activity of control plasma membrane vesicles from that of ABCG2-expressing plasma membrane vesicles.

### 4.11. Statistical Analysis

All statistical analyses were performed by using EXCEL 2019 (Microsoft, Redmond, WA, USA) with Statcel4 add-in software (OMS publishing, Saitama, Japan). The number of biological replicates (*n*) is described in the figure legends. In single pairs of quantitative data, after comparing the variances of a data set (using the *F*-test), an unpaired Student’s *t*-test was performed. Statistical significance was defined in terms of *P* values less than 0.05 or 0.01.

## Figures and Tables

**Figure 1 ijms-22-01935-f001:**
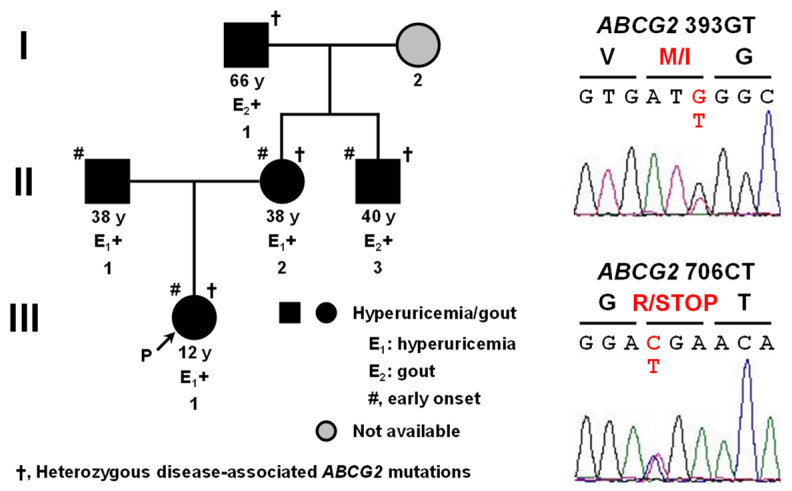
Identification of novel disease-associated mutations c.393G>T (p.M131I) and c.706C>T (p.R236X) in *ABCG2*. Left, the pedigree of a Czech family with early-onset hyperuricemia and gout; Right, representative electropherograms of partial sequences of *ABCG2* showing the heterozygous point mutations (red) discovered in our present study. I–III, generation of the family; P, proband; y, years old.

**Figure 2 ijms-22-01935-f002:**
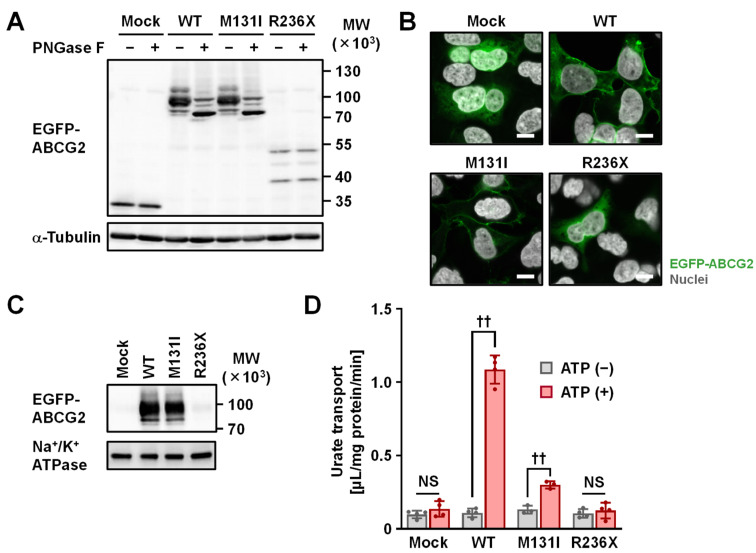
Functional characterization of disease-associated ABCG2 mutations. (**A**) Immunoblot of whole cell lysate samples. α-Tubulin used as the loading control. WT, wild-type. (**B**) Confocal microscopy of intracellular localization. Nuclei were stained with TO-PRO-3 iodide (gray). Bars, 10 μm. (**C**) Immunoblot of plasma membrane vesicles. Na^+^/K^+^ ATPase as the loading control. (**D**) Urate transport activity. The incubation condition was 37 °C for 10 min in the presence of 20 μM radiolabeled urate. Data are expressed as the mean ± SD. *n* = 3–4. ††, *P* < 0.01; NS, not significantly different between groups (two-sided *t*-test).

**Figure 3 ijms-22-01935-f003:**
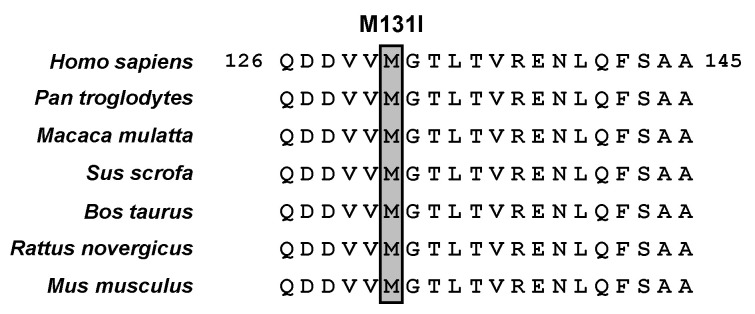
ABCG2 M131 is shown to be evolutionary conserved among seven mammalian species. Regarding the Abcg2 protein in each species, the NCBI Reference Sequence IDs are summarized as *Pan troglodytes* (Chimpanzee), GABE01009237.1; *Macaca mulatta* (Rhesus macaque), NM_001032919.1; *Sus scrofa* (Pig), NM_214010.1; *Bos taurus* (Bovine), NM_001037478.3; *Rattus norvegicus* (Rat), NM_181381.2; *Mus musculus* (Mouse), NM_011920.3. Multiple sequence alignments and homology calculations were carried out using GENETYX software (GENETYX, Tokyo, Japan) and the ClustalW2.1 Windows program, per the protocol used in our previous study [40].

**Table 1 ijms-22-01935-t001:** Clinical and biochemical data of a family with early-onset hyperuricemia and gout.

	Proband	Mother	Maternal uncle	Maternal grandfather	Father
**Age of onset (HA/gout)**	12 years (HA)	35 years (HA)	35 years (gout)	43 years (gout)	30–35 (HA)
**Main symptoms**	Asymptomatic	Asymptomatic	Gout	Gout	Asymptomatic
**Other symptoms or diseases**	Astigmatism; Bilateral cataracts; Myopia; Bronchial asthma; Obesity	Thyropathy; Vertebrogenic Algic syndrome	T2D; Hypertension; Obesity; Thyropathy	T2D; Hypertension	Bronchial asthma; Hypertriglyceridemia; Hypertension; Central obesity
**SUA before treatment [µmol/L (mg/dL)]**	397–405 (6.67–6.81)	420–439 (7.06–7.38)	>500 (>8.41)	537 (9.03)	487 (8.19)
**UUA** **[mmol/mol creatinine]**	3.34	1.47	2.10	n.d.	2.45
**FE-UA [%]**	2.2	4.6	2.9	n.d.	3.9
**Therapy**	No	No	Allopurinol (100 mg per day)	Colchicine; Allopurinol (300 mg per day)	No
**SUA during treatment [µmol/L (mg/dL)]**	n.d.	n.d.	446 (7.50) (non-compliance)	422 (7.09) (non-compliance)	n.d.

HA, hyperuricemia; SUA, serum uric acid; UUA, urine uric acid; FE-UA, fractional excretion of uric acid; T2D, type 2 diabetes mellitus; n.d., not determined. Reference ranges are as follows. SUA: 120–360 μmol/L (2.02–6.05 mg/dL) (< 15 years and female), 120–420 μmol/L (2.02–7.06 mg/dL) (male); UUA: 0.1–1.0 mmol/mol creatinine (< 15 years), 0.1–0.8 mmol/mol creatinine (≥15 years); FE-UA: 5–20% (< 13 years), 5–12% (male), 5–15% (female).

## Data Availability

Data supporting the findings of this study are included in this published article or are available from the corresponding author on reasonable request.

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
