# Peer review of "Identification of Two Dysfunctional Variants in the ABCG2 Urate Transporter Associated with Pediatric-Onset of Familial Hyperuricemia and Early-Onset Gout"

_ijms, 2021, doi:10.3390/ijms22041935_

Round 1

Reviewer 1 Report

In general, I think this was a very clear and concise manuscript. I have only a few comments:

In figure 2D some more details on the data could be given which is now only found in the methods. Fo instance the tested concentration, time etc.

In addition, I would like the authors to add their motivation to test only one urate concentration and why that specific concentration was chosen. I realize that determining Km or Vmax is much more work, but  at least for the M131I variant could be relevant.

M131 appears to belong to the q-loop motif of the NBD, which in turn suggest that the mutant affect the ATPase catalytic cycle, this might be added to the discussion.

Has the functionality and expression of the  wt EGFP-ABCG2 fusion been compared to the nonfusion wtABCG2? I maybe missed a reference to a previous study where this has been done. If not present, I think it should be added, as it is quite important.

Reviewer 2 Report

This is a very interesting description of a case of interaction between genetic mutations in a transporter and hyperuricemia. The results are well supported and, even it is a mere case, the interest is high as a support for further research.

Reviewer 3 Report

In this work, Toyoda and colleagues studied the effect of single nucleotide polymorphisms of the ABC transporter ABCG2 observed in a young girl and her family members with premature hyperuricemia on the expression and uric acid efflux behaviour of the transporter. In a “from bedside to bench” approach, these were found to be dysfunctional variants of the transporter, one associated with the formation of a truncated protein.

Abstract

In the abstract and also in the introduction in the results section (Table 1) of the manuscript, the common units for uric acid (µmol/L and mg/dL) should both be taken into account, so that the data can be more easily grasped by clinicians of different countries. In addition, the normal range of serum uric acid in children should be discussed and how the values of the investigated girl deviate here. Also, the normal range for adults should be related to uric acid values in children in the body of the manuscript.

Introduction:

Page 2, lines 45-48: since this section is of great contextual importance for understanding hyperuricemia and gout in younger age, it should be more detailed and accurately address the specifics of "early onset gout."

Results

Figure 2B-D: the structure of the EGFP-ABCG2 fusion construct should be given in a supplemental file. Were the vesicles for the uric acid transport study obtained from cells expressing EGFP-ABCG2 or the variants? It should then be shown here that EGFP does not alter the efflux behavior of the transporter compared with the unlabeled wild type.

Figure 2D: the number of replicates is too low. Please increase to n=5 at least.

Discussion

The impact of the M131I variant on uric acid (substrate) transport and induced ATPase activity should be evaluated and discussed in more detail in the light of recent primary or secondary literature (Nature. 2018 Nov;563(7731):426-430; Br J Pharmacol. 2020 Apr;177(7):1485-1496). What about its effect on the ABC signature function?

Lines 243-256: the findings of the manuscript should also be discussed in light of recent interesting work on urate secretion by ABCG2 (intestinal portion; Nat Commun. 2020 Jun 2;11(1):2767). 

Material and Methods

Clinical Subjects:

The authors should confirm that all studies involving human subjects were conducted in accordance with the Ethical Principles for Medical Research Involving Human Subjects as defined in the Declaration of Helsinki.

Is there a registration number for the clinical investigations performed in connection with the case described in the manuscript?

Round 2

Reviewer 1 Report

The authors responses to my comments were sufficient.